# Sensitive, Color-Indicating and Labeling-Free Multi-Detection Cholesteric Liquid Crystal Biosensing Chips for Detecting Albumin

**DOI:** 10.3390/polym13091463

**Published:** 2021-05-01

**Authors:** I-Te Wang, Yen-Hua Lee, Er-Yuan Chuang, Yu-Cheng Hsiao

**Affiliations:** 1Department of Obstetrics and Gynecology, Taipei Medical University Hospital, Taipei Medical University, Taipei 11031, Taiwan; b8301120@yahoo.com.tw; 2Graduate Institute of Biomedical Optomechatronics, College of Biomedical Engineering, Taipei 11031, Taiwan; yhlee@tmu.edu.tw; 3Graduate Institute of Biomedical Materials and Tissue Engineering, College of Biomedical Engineering, Taipei Medical University, Taipei 11031, Taiwan; 4International PhD Program for Biomedical Engineering, Taipei Medical University, Taipei 11031, Taiwan; 5Cell Physiology and Molecular Image Research Center, Wan Fang Hospital, Taipei Medical University, Taipei 11031, Taiwan; 6Stanford Byers Center for Biodesign, Stanford, CA 94305-5428, USA

**Keywords:** cholesteric liquid crystal, microfluidic chip, BSA

## Abstract

A novel device for cholesteric liquid crystal (CLC)-based microfluidic chips, accommodated in a polydimethylsiloxane material, was invented. In this device, the reorientation of the CLCs was consistently influenced by the surface of the four channel walls and adjacent CLCs. When the inside of the microchannel was coated with the alignment layer, the CLCs oriented homeotropically in a focal conic state under cross-polarizers. Once antigens had bound onto antibodies immobilized onto the orientation sheet-coated channel walls, the light intensity of the CLC molecules converted from a focal conic state to a bright planar state caused by disrupting the CLCs. By means of utilizing pressure-propelling flow, the attachment of antigen/antibody to the CLCs should be detectable within consecutive sequences. The multi-microfluidic CLC-based chips were verified by measuring bovine serum albumin (BSA) and immune complexes of pairs of BSA antigen/antibody. We showed that the multiple microfluidic immunoassaying can be used for measuring BSA and pairs of antigen/antibody BSA with a detection limit of about 1 ng/mL. The linear range is 0.1 μg/mL–1 mg/mL. A limit of immune detection of pairs of BSA antigens/antibodies was 10 ng/mL of BSA plus 1000 ng/mL of the anti-BSA antibodies was observed. According to this innovative creation of immunoassaying, an unsophisticated multi-detection device with CLC-based labeling-free microfluidic chips is presented.

## 1. Introduction

Due to fast detection, small sample volume and cheap price, microfluidic chips have been wildly applied [1,2]. However, the signal of microfluidic immunoassay is weak because of the microscale sample. In order to make the antigen and antibody binding become a detectable signal, the antigen or antibody is usually labeled by an fluorophores [3,4] enzyme [5,6], or nanoparticles [7,8]. However, when an antigen or antibody is conjugated with a label, the binding affinity might be affected when transducing the immune-binding response to measurable signs. In addition, pairs of antibody/antigen are influenced by being conjugated with a label [9,10]. The most important is the detection of standard bovine serum albumin (BSA) for the observation and establishing of characteristics. The current research of BSA immunoassay detection involves labeling by fluorophores [3,4] for biochemical analysis.

Recently, label-free liquid crystal (LC) biosensors have been developed. The immunobinding responses are able to re-orient the LC molecules and change their optical signals. The optical property changes of LCs enable naked-eye detection of label-free immunoassays [11]. It was stated that the re-orientating LCs provide sensitivity to immunobinding responses and change the optical signals of LCs [12,13]. A previous study combined LC with microfluidic devices to detect ethanol and bovine serum albumin (BSA) [14,15]. In addition to nematic LCs, cholesteric LCs (CLCs) have unique optical properties like Bragg reflection, bi-stability and flexibility [16,17,18,19,20,21]. The first biologically used CLC sensor device was created by our team in 2015 [22]. A high-sensitivity color-specifying CLC biosensor has been conceived. Nevertheless, the CLC biosensors need complicated fabrication processes and have to be restricted inside a stated area, for instance, a transmission electron microscopic grid [23] or a cell device [24]. To simplify the processes, a single-substrate device has been invented [25]. Furthermore, CLC biosensors can be integrate with a smartphone so it is possible to detect various disease biomarkers at home. Due to the aforementioned advantages, we proposed to integrate CLC materials with microfluidic devices and to investigate the behavior between CLCs and pairs of BSA antigens/antibodies in microchannels [26]. That expedient should enable observing the formation of organic ethanol, but it is just as valid for measuring organic substances. In addition, Fan et al. reported an LC microfluidic system for measuring bimolecular BSA. A robust, unsophisticated appliance for an LC-based microfluidic immunodetecting method has been proposed [27]. The most important related technology of these label-free sensors are grating coupled interferometry (GCI) and plasmonic sensing [28,29,30,31]. However, compared to plasmonic sensing or GCI, LC biosensors are more portable, cheaper and can be monitored with the naked eye [26,27].

In this paper, we present a CLC-based multiple microfluidic biosensing chip. The behavior between pairs of BSA antigens/antibodies and molecules of CLC in a microchannel was investigated. Our team suggests that this CLC-based multiple microfluidic biosensing chip substantially differed from a typical biosensor. The alignments of the CLCs have been ascertained with both the functionality of the interface and the dimensions of geometry of the channel. The antigens/antibodies were able to be measured by studying the optical characteristics of the device of CLCs within the microchannel below cross-polarizers. A very delicate interface among molecules of CLC and an arrangement layer of N,N-dimethyl-n-octadecyl-3-aminopropyltrimethoxysilyl chloride (DMOAP) were used for measuring concentrations of existing BSA. A schematically illustrated figure of this multiple microfluidic CLC biological sensor is shown in Figure 1.

## 2. Materials and Methods

To generate the two levels with single-layer cascading microchannels, a microchannel mold of thickness 25 μm was made on a silicon wafer (4 inch) through a polydimethylsiloxane (PDMS)-based soft lithographical manufacturing procedure using photoresist (SU-8 2025). The PDMS base was mixed with the curing agent (one-tenth of the PDMS weight) and degassing for 30 min. Then, the mixture was poured onto a master and baked for 4 h at 65 °C. Afterwards, the PDMS was shed off from the master and pre-cleansed glass was attached tightly together by using oxygen plasma treatment. The LC E7 (Merck, Darmstadt, Germany) mixed with chiral dopant R5011 (Merck, Darmstadt, Germany) was applied in this investigation. To coat the aligned layer, the multiple microfluidic channels were put under the DMOAP fluid for 0.5 h. Then, the coated microfluidic channels were washed with water for one minute. For the immobilization test, the BSA solution (0~1 mg/mL) or anti-BSA antibody (0~1000 µg/mL) was packed into the DMOAP-coated channel of the microfluidic device. To make CLC multi-microfluidic chips, the CLCs were poured into empty microfluidic channels with a syringe under different volume flow rates (5, 10, 20 and 30 µL/min).

## 3. Results and Discussion

### 3.1. Detecting BSA by the Multi-Microfluidic CLC Biosensors

The design of the CLC microfluidic biosensors is illustrated in Figure 1. The anti-BSA and BSA were loaded into the CLC microfluidic biosensor, as shown in Figure 1b,c. Finally, the LC was injected into the microfluidic device. The polarized optical image under a cross-polarizer and the optical mechanism of the CLC biosensor are also shown in Figure 2. The different colors can be observed ranging from 0–1 mg/mL of BSA. Since the vertically aligned DMOAP layer made the CLCs orient perpendicularly toward the interface of the channel of the microfluidic device, it appeared dark and deprived of biomolecules. Once the vertical anchoring force was diminished by BSA, the focal conic (FC) state near the microfluidic channel changed to a planar (P) state. Note that FC is the scattering multi-domain of the LC and P is the bright helical LC structure. Although the change of color could be observed without any polarizer, the contrast under a cross-polarizer was better. Thus, we used a cross-polarizer to quantify the data results of CLC biosensors. The CLC microfluidic biological sensor should be applied along an extensive range of temperatures in various applications. To test the relation between BSA concentration and the image of the CLC biosensor, 0~1 mg/mL BSA was added to the CLC microfluidic biosensor and the polarized images are shown in Figure 3. The CLC microfluidic biosensor was blue when there was no of BSA, and it shifted to green with an increasing BSA concentration. This result indicated that the multiple microfluidic CLC chips device can be utilized for measuring BSA concentrations. In order to quantify the results of the CLC biological sensor, the intensity of the image was analyzed by using the software ImageJ [32]. The result is shown in Figure 4. The intensity of CLC chips and the logarithm of the BSA concentration showed a linear correlation. Therefore, the CLC microfluidic biological sensor could be used to measure the concentration of BSA successfully. A linear relationship with a determination coefficient R^2^ = 0.985 was found.

### 3.2. Combining BSA Antibody with the CLC Microfluidic Device

In order to use this BSA CLC microfluidic device in clinical applications, anti-BSA antibody (0, 10, 100 and 1000 µg/mL) was immobilized on the CLC microfluidic device before adding the BSA samples. The results show that a CLC chip can also be tested by means of BSA with anti-BSA antibodies. The optical level of the immune assaying CLC microfluidic device was halted by 0~10 μg/mL of BSA and 0 to 1 mg/mL of the anti-BSA antibodies, as shown in Figure 5. The concentration of 0 to 1 mg/mL of the anti-BSA antibody with the BSA antigens (0 to 10 μg/mL) was used for allowing the generation of immune complexes among pairs of particular antigens/antibodies. The smaller amount of the anti-BSA antibodies of below 10 µg/mL was incapable of making immune complexes of given antigens/antibodies. The amount of immunocomplexes with 1 and 10 μg/mL of BSA did not changes. Once 0.1 and 1 mg/mL concentrations of the anti-BSA antibodies were blended, the immune complexes led to a much brighter state. Excess amounts of the anti-BSA antibodies will change the orientations of CLCs, inducing high brightness. Data showed that immune complexes of BSA, in comparison with those with the BSA antigens or antibodies alone, produced more meaningful disturbances of the CLC orientations (Figure 5). Therefore, 1 µg/mL of the anti-BSA antibody provided an appropriate interruption with the antigen of BSA. The CLC microfluidic device chip could discern immune complexes and unchained antibodies with antigens. In addition, the proportional relationship between the amount of CLC-based microfluidic device chips and BSA molecules within the anti-BSA antibodies is revealed in Figure 6. The limit of immune detection of pairs of BSA antigens/antibodies was 10 ng/mL of BSA plus 1000 ng/mL of the anti-BSA antibodies. These data show that the proportional relationship of the CLC-based multiple microfluidic devices should be quantitatively created for immune detection in a linear way. A linear relationship with a determination coefficient R^2^ = 0.9 was found. In comparison to the well-recognized immune detection method, our multiple microfluidic LC device immune assaying chips are inexpensive and easier to use. According to the naked-eye observation of the nature of label-free devices, the investigation suggests that the LC microfluidic device has potential for advancement as an inexpensive, small and portable biological sensing method for immune detection.

### 3.3. Volume Flow Rate Effects of CLC Microfluidic Chips

Volume flow rates of fluids of CLCs into microfluidic chips were investigated. The results of Figure 7 suggest the consequence of various volume flow rates of CLCs into the channels of chip. A rapid volume flow rate (over 0.01 mL/min) caused a disarranged shape of CLCs and produced an imperfect texture [33]. Once the flow rate of volume was over 0.03 mL/min, the overly rapid volume flow rate caused a small amount of CLC to be lost in the microfluid channel. Due to the disordered orientation of CLCs under rapid volume flow rates, the optical density of the microfluid channels was lower when the flow rate of volume increased. We employed a flow rate of volume of below 0.01 mL/min based on these results.

## 4. Conclusions

CLC-based microfluidic biological sensing chip devices were shown in this investigation. The orientated CLCs were influenced by the surface of the four microchannel walls. The alignment of the DMOAP layer was on the interior of the microchannel, and the CLCs aligned perpendicularly and displayed a blue color reflected under cross-polarizers. Once the antigens of BSA bound onto the BSA [34] antibodies immobilized in the device, the CLC phase color switched from blue to green owing to interruption of the orientation of CLC. The linear range is 0.1–1 mg/mL. By means of pressurized flow, BSA antigen/antibody attachment should be measured with polarized optical microscopy. Moreover, the immune detection limit of BSA antigens/antibodies was 10 ng/mL of BSA and 1000 ng/mL of the anti-BSA antibodies in the CLC device. This suggested that this multiple microfluidic CLC immune assaying chip device could measure BSA and antigens/antibodies of BSA immune complexes with label-free immune detection. The innovative development of this immune assaying device offers a precise, economical, multiple detection, color-specifying and vigorous method for CLC-based immune detection. Based on the results of the CLC biosensor, we may be able to change it for use with anti-SARS-CoV-2 antibodies to capture SARS-CoV-2 and affect the arrangement of CLC as a new sensor. This may have great advantages for coronavirus disease quarantine.

## Figures and Tables

**Figure 1 polymers-13-01463-f001:**
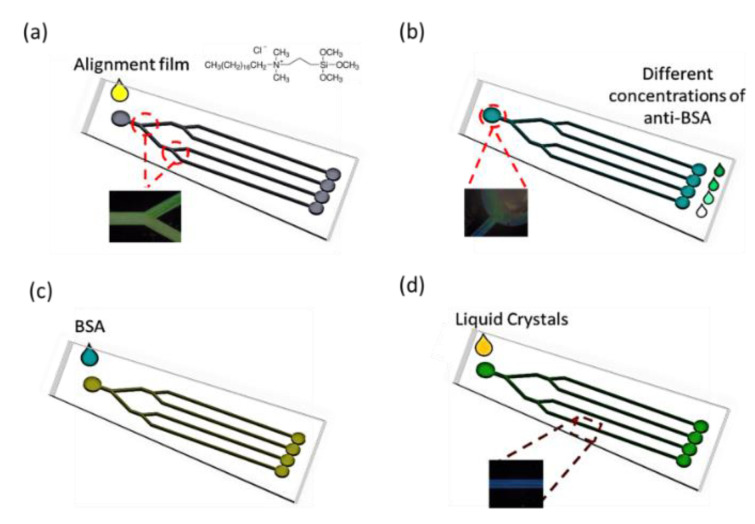
Illustration of microfluidic cholesteric liquid crystal (CLC) biological sensor chip in the presence of bovine serum albumin (BSA) biomolecules in DMOAP–coated microfluidic channels. (**a**) drop the alignment (**b**) drop the anti-BSA (**c**) drop the BSA (**d**) infiltrate the liquid crystals.

**Figure 2 polymers-13-01463-f002:**
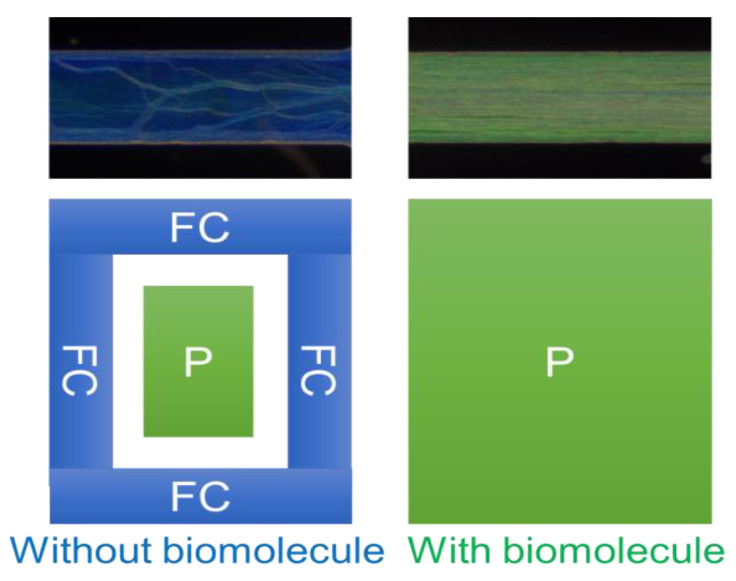
Data from polarized optical (light) microscopy of CLC multiple microfluidic biological sensors with concentrations of BSA ranging from 0–1 mg/mL under conditions of a cross-polarizer. The ophthalmic mechanism of the CLC biological sensor both without and with BSA is shown.

**Figure 3 polymers-13-01463-f003:**
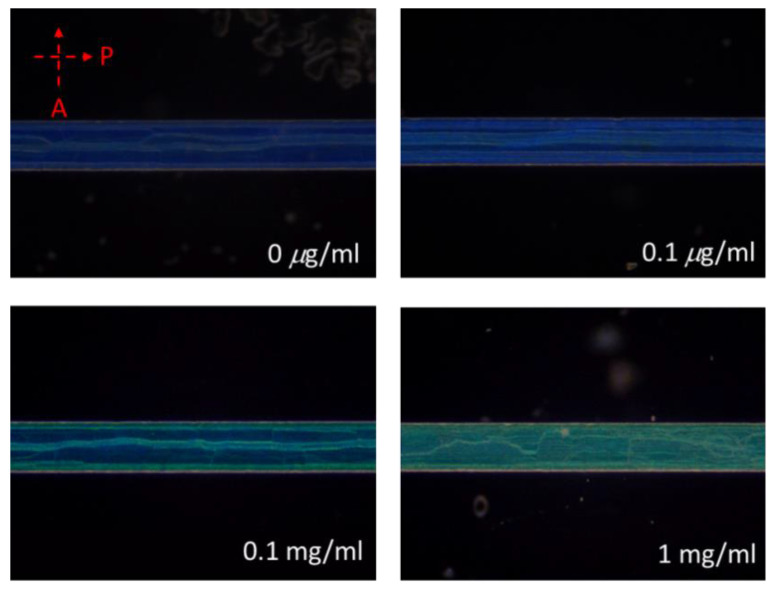
Polarized optical images of cholesteric liquid crystal (CLC) microfluidic biological sensors with immobilized BSA ranging from 0 to 1 mg/mL.

**Figure 4 polymers-13-01463-f004:**
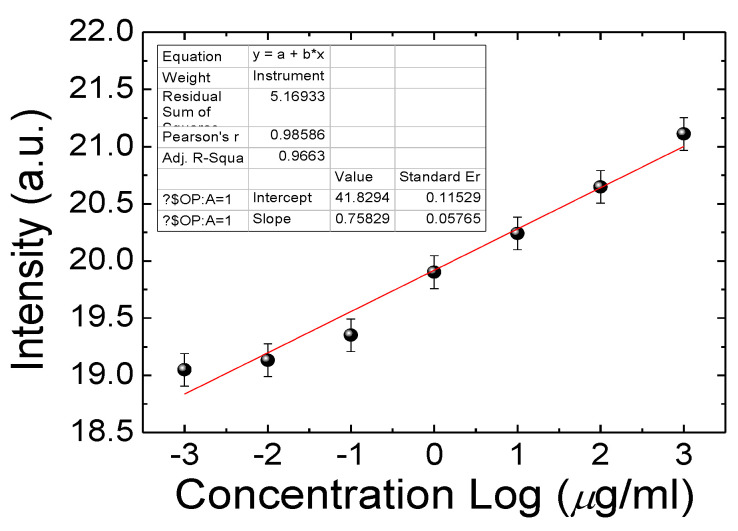
Proportional relationship of the transmitted intensity of cholesteric liquid crystal (CLC) multi-microfluidic chips at various concentrations of BSA.

**Figure 5 polymers-13-01463-f005:**
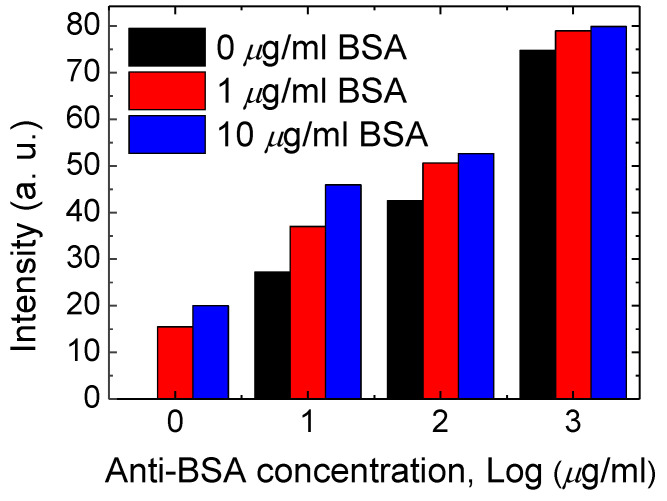
Intensity of immune assaying CLC chips immobilized by concentrations ranging from 0 to 0.01 mg/mL of BSA and 0 to 1 mg/mL of the anti-BSA antibodies.

**Figure 6 polymers-13-01463-f006:**
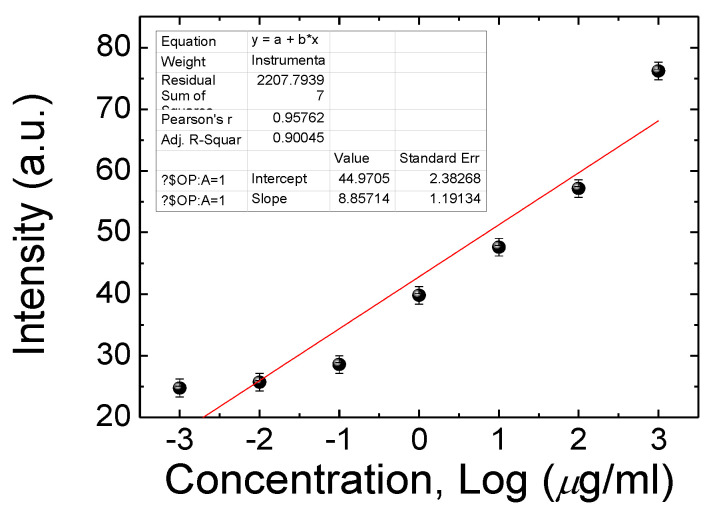
Photo intensities of CLC immune assaying chip devices at a various concentrations of BSA with 0.01 mg/mL of anti-BSA antibodies.

**Figure 7 polymers-13-01463-f007:**
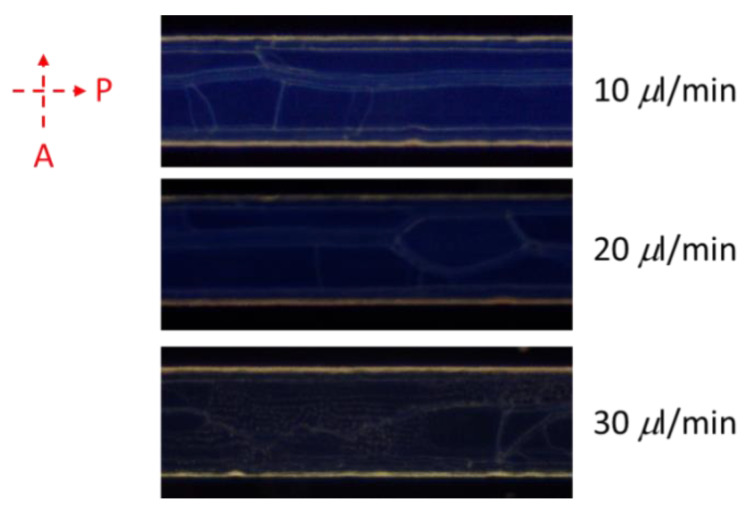
Image data of a 1 mg/mL of BSA under various volume flow rates of CLCs into the microchannel.

## Data Availability

The data presented in this study are available on request from the corresponding author.

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
