# Peer review of "Sensitive, Color-Indicating and Labeling-Free Multi-Detection Cholesteric Liquid Crystal Biosensing Chips for Detecting Albumin"

_polymers, 2021, doi:10.3390/polym13091463_

Round 1

Reviewer 1 Report

Comment:

Tiple micro fluidic CLC immune-assaying chip device was used to measure BSA and antigens/antibodies of BSA. The innovative development of this immune-assaying device offers a precise, economy, multiple detection, color-specifying, and vigorous way for CLC-constructed immune-detection.The manuscript needs to be modified according to the following suggestions:

  1. Please provide a linear range that can be quantitatively detected.
  2. line 126 “10 μg/ml”should be 10 μg/mL. These similar problems appear many times in the manuscript, please correct.
  3. Please provide linear correlation coefficient and linear equation (Figure 4 and Figure 6).
  4. Please clarify the current research status of BSA and the pairs of anti-gen/anti-body BSA detection methods in Introduction.

Reviewer 2 Report

The article under review consists of the development of cholesteric sensitive liquid crystal biodetection chips, with color indication and without labeling for the detection of albumin. The study presented here is the continuation of work initiated in 2015 by the authors and developped in 2020 in Polymers.

The novelty of this study compared to the last published work of the authors is not sufficiently explained; this must be clearly explained and the innovative character put more forward.

I therefore propose major revisions before publication.

Minor and Major Recommandations;

- Abstract should be reviewed as it does not contain the objectives and results of the study presented in this article.

- Keywords listed by the authors are not relevant to the article.

- Improve Introduction part; no sufficient background and so include all relevant references

- Discussion part must be developed by more comments and conclusions because of all the results obtained.

- line 78; "The LC E7 mixed with 78 chiral-dopant-R5011...." origin, supplier, ....?

- In Figure2; re-define focal conic (FC) and planar (P) state

- Why a bad correlation in Figure 6

- Harmonize in the text  "Fig" or "Figure"

- More than 10% self-citation. Include more recent references.

Round 2

Reviewer 1 Report

accept

Reviewer 2 Report

The revised version of the manuscript was improved and now warrants publication in Polymers.